# A New Unsupervised Technique to Analyze the Centroid and Frequency of Keyphrases from Academic Articles

**Mohammad Badrul Alam Miah** [1,2], **Suryanti Awang** [1,3,*], **Md Mustafizur Rahman** [4], **A. S. M. Sanwar Hosen** [5] **and In-Ho Ra** [6,*]

1 Faculty of Computing, Universiti Malaysia Pahang, Pekan 26600, Malaysia
2 Department of Information and Communication Technology, Mawlana Bhashani Science and Technology University, Tangail 1902, Bangladesh
3 Center of Excellence for Artificial Intelligence & Data Science, Universiti Malaysia Pahang, Lebuhraya Tun Razak, Gambang 26300, Malaysia
4 Department of Mechanical Engineering, Faculty of Engineering, Universiti Malaysia Pahang, Gambang 26300, Malaysia
5 Division of Computer Science and Engineering, Jeonbuk National University, Jeonju 54896, Korea
6 School of Computer, Information and Communication Engineering, Kunsan National University, Gunsan 54150, Korea
* Correspondence: suryanti@ump.edu.my (S.A.); ihra@kunsan.ac.kr (I.-H.R.)

**Abstract:** Automated keyphrase extraction is crucial for extracting and summarizing relevant information from a variety of publications in multiple domains. However, the extraction of good-quality keyphrases and the summarising of information to a good standard have become extremely challenging in recent research because of the advancement of technology and the exponential development of digital sources and textual information. Because of this, the usage of keyphrase features for keyphrase extraction techniques has recently gained tremendous popularity. This paper proposed a new unsupervised region-based keyphrase centroid and frequency analysis technique, named the KCFA technique, for keyphrase extraction as a feature. Data/datasets collection, data pre-processing, statistical methodologies, curve plotting analysis, and curve fitting technique are the five main processes in the proposed technique. To begin, the technique collects multiple datasets from diverse sources, which are then input into the data pre-processing step by utilizing some text pre-processing processes. Afterward, the region-based statistical methodologies receive the pre-processed data, followed by the curve plotting examination and, lastly, the curve fitting technique. The proposed technique is then tested and evaluated using ten (10) best-accessible benchmark datasets from various disciplines. The proposed approach is then compared to our available methods to demonstrate its efficacy, advantages, and importance. Lastly, the results of the experiment show that the proposed method works well to analyze the centroid and frequency of keyphrases from academic articles. It provides a centroid of 706.66 and a frequency of 38.95% in the first region, 2454.21 and 7.98% in the second region, for a total frequency of 68.11%.

**Keywords:** keyphrase extraction; KCFA technique; data pre-processing; curve plotting; curve fitting technique; feature; keyphrase centroid; keyphrase frequency

## 1. Introduction

The exponential growth of textual content, as well as the ongoing expansion of the information age, makes handling such a massive amount of data much more challenging. Online textual content is either semi-structured or unstructured; examples include academic papers, online journals, news sources, and books [1]. Prior to the development of technology, people could only process this large amount of data, which took a long time [2]. Furthermore, it is difficult to accomplish this massive amount of data because

of discrepancies between the quantity of information and manual information process skills, leading to the development of automated keyphrase extraction techniques that use computers' comprehensive computational power to replace physical labor [3,4].

Automatic keyphrase extraction methods are used to extract high-level key phrases from articles. In general, the keyphrase provides a high degree of document characterisation, summary, and description, which is important for a number of Natural Language Processing (NLP) features such as article classification, clustering, as well as categorization [5,6]. "Despite this, they are employed in a broad range of Digital Information Processing applications including Information Retrieval, Digital Content Management, Recommender Systems [7], and Contextual Advertising" [2]. Among other things, it can be used for search engines, media searches, legal and geographic information retrieval, and digital libraries [2,8,9].

Several keyword/keyphrase extraction approaches have been devised to fit the above-mentioned applications [10–15]. Among them, domain-specific tactics [10,16] necessitate application domain expertise; linguistic approaches [11] necessitate language proficiency. As a result, they are unable to tackle problems in other subjects/domains or languages. Supervised machine learning (SML) algorithms take a substantial amount of rare training dataset to extract quality keystones and generalize poorly outside of the scope of training data, according to [17,18]. It also increased the storage and computation, decreased the comprehensibility, and made the system computationally expensive [3,19,20]. Again, because of the huge number of complex processes, statistical unsupervised techniques such as [15,21] are computationally expensive. Graph-based unsupervised approaches perform badly because of their inability to detect cohesion amongst numerous words that compose a keyphrase [22–27]. Finally, TeKET [14] is extremely versatile and acts similarly to TF-IDF for short data lengths.

Keyphrase centroid and frequency analysis (KCFA) is required for those keyphrase extraction techniques that extract high-level keyphrases from articles. To identify keywords from other phrases, the proposed KCFA approach can be utilized as a characteristic/feature of such keyphrase extractions. The keyword extraction technique cacannot extract quality keywords without using good quality features [28]. It has been established that, as a result of the prior debate, keyphrases feature extraction remains a critical research area for the survey.

As a result, this paper provides an unsupervised new KCFA technique with the following notable contributions:

- the proposed approach is corpus, domain, and language agnostic;
- both supervised and unsupervised techniques can be benefited from the proposed technique;
- the proposed method is a document-length-agnostic approach;
- ten (10) standard datasets were utilized to analyze and evaluate the efficiency of the suggested technique.

The following is the structure of the rest of this article: The numerous techniques are outlined in Section 2, together with their strengths and limitations, emphasizing the necessity for a new technique to still be offered. Afterwards, in Section 3, a novel region-based unsupervised KCFA technique is described for determining the keyphrase centroid and frequency in each region of an article. Afterward, in Section 4, the experiments' setup is described in depth, that includes datasets details, evaluation metrics, as well as implementation details. Then, the proposed technique was then examined on ten (10) standard datasets and evaluated for the effectiveness of the system, and then compared to current methods to determine their benefits and drawbacks, which are seen in detail in Section 5. Finally, the study's contributions, future works, and shortcomings would be identified and stated in Section 6.

## 2. Background Study

The proposed strategy is a new method for analyzing keyphrases centroid and frequency from articles which may be utilized as a characteristic/feature for key extraction technique. Hence, the section covers comparable techniques. Based on the training datasets,

There are two sorts of key extraction techniques: unsupervised and supervised [2,29]. Both approaches make use of features and feature extraction techniques. Here, we go through the key points of both parties' techniques in the following subsections.

### 2.1. Unsupervised Techniques

Utilizing this technique, the keyphrase extraction approach is a ranked problem that can be addressed without any prior experience. These methods are categorized as graph-based or statistical-based, according to [30]. The parts that follow go into great detail about the most essential techniques utilized by both groups.

PageRank is indeed a graph-based method that is built on random walks. It is fine for sifting through web pages and social media pages, but it cacannot extract crucial information from authorized manuscripts [17,31]. The PositionRank is the extension of PageRank that has been established to improve performance, and it evaluates words by considering all of their placements and frequency, determining their rank. However, because it overlooks thematic coverage and diversity, this method performs badly [26].

TextRank employs POS tag like an intrinsic feature, but it has a number of drawbacks, including the difficulty to capture cohesion, which leads to sub-optimal outcomes [23]. Another major extraction technique that overcomes TextRank's restrictions is TopicRank. TopicRank extracts noun phrases from the document and groups them into subjects. It also has a problem with error propagation [24]. TextRank's lengthening is SingleRank. By acquiring ranked words, it accurately extracts just noun phrases from datasets, not keyphrases. In ranking phase, unimportant keywords are used, although this does not always screen out small scoring terms, providing longer keywords greater scores [22].

The TopicRank propagation matter is resolved using the MultipartiteRank technique. However, it has a clustering inaccuracy, making it difficult to choose the most relevant candidates [27]. The well-known unsupervised graph-based keyword extraction method is named TeKET (Tree-based Keyphrase Extraction Technique), which is domain and language agnostic, and requires fundamental statistical understanding. Although this approach beats the several important keyphrase extraction strategies, it includes some drawbacks, such as providing extensive flexibility [14].

The Term Frequency–Inverse Document Frequency (TF-IDF) is the most extensively utilized statistical technique. Though TF-IDF technique is straightforward to build, determining the IDF with a big dataset needs a lengthy time as well as lots of computer resources [32]. The KP-Miner algorithm is also employed to resolve the single-term preferences issues. Though it surpasses TF-IDF, which has a number of flaws, such as a declining world ranking performance as data rises. Since it depends on TF-IDF, it also is computationally costly [29].

"Yet Another Keyword Extractor (YAKE) calculates the weighting scores of a keyword utilizing five key elements: casing, term relatedness to context, term position, term distinct sentence, and term frequency normalization" [2]. Furthermore, since it generates candidate keys using the N-grams approach, its computational cost effect increases with this N-grams technique [15].

### 2.2. Supervised Techniques

Using this technique from the article, the keyword extraction methodology is considered a binary category problem, with a fraction of the candidate keywords classified as either keywords or non-keywords. Neural networks (NN) [33,34], Nave Bayes, Support vector machines (SVM), decision trees (DT), and C4.5 [2,4] are some of the approaches that may be used to solve the classification problem. The most important techniques are reviewed in depth in the following that uses this approach.

Keyphrase Extraction Algorithm (KEA) makes advantage of TFxIDF and first occurrence position as a feature [30,35]. It employs illustrative methodologies for detecting candidate keyphrases, for each candidate estimating feature values, and utilizing the Naive Bayes algorithm for predicting and determining candidates' quality keyphrases. However,

because KEA is dependent on the training dataset, it could give poor results if the training dataset does not match the document. Genitor Extractor (GenEx) automatically takes first occurrence position, keyphrase length, and term frequency (TF) as a feature [30,36]. The most extensively used keyword extraction method is based on a C4.5 decision-making approach that involves genetic algorithms to maintain its efficiency across domains. This system does not employ the TFxIDF technique.

The Hulth system permits the retrieved keyphrases to be as lengthy as they wish to be, in contrast to the KEA and GenEx approaches [30]. "N-grams, Part-of-speech tag known as POS-tag, first occurrence position, TF, and noun phrase (NP) are the four properties it uses" [2]. Regrettably, there is no correlation between of different POS-tag attributes. The technique does not try against the GenEx/KEA criteria, and it reports that the value of recall is low.

The Maui Algorithm is an automated generalized topical indexing method which is based upon that KEA system [30,37]. It expands the KEA system by including data from Wikipedia. However, one of this algorithm's shortcomings is that it lacks assessment capabilities.

Automatic Key Term Extraction from Scientific Articles known as (HUMB), uses the location of a term with its initial occurrence; phraseness; informativeness; keywordness; and the candidate term's length as a feature [30,38]. It has shown positive outcomes in a range of data sets. It, on the other hand, relied on external knowledge sources General Research Insight in Scientific and technical Publications (GRISP), Hyper Article en Ligne (HAL) and GeneRation Of BIbilographic Data (GROBID)/Text Encoding Initiative (TEI)) that is linked to scientific disciplines.

The Document Phrase Maximality (DPM) index uses a total of eighteen (18) statistical factors [39]. The DPM index and an additional five (5) are unique features amongst them. Compared to other keyphrase extraction methods, this system's outcomes have improved dramatically without utilizing outer knowledge or manuscript structural elements.

Citation-enhanced keyphrase extraction is a supervised model known as CeKE [40]. The following essential features are used by the CeKE: Relative position, TFxIDF, POS tag, inCited and inCiting, citation TF-IDF, TF-IDF-Over, firstPosUnder, first position. They have the ability to improve keywords extraction and add important features. In comparison to previous systems, the CeKE+ keyness model produces noteworthy results [41].

Using supervised learning approaches, the Keyphrase Extraction (KeyEx) Method identifies a huge number of probable candidate keys and builds a classifier standard for keyphrase extraction [42]. Experiments by the author revealed that the KeyEx method considerably enhanced the quality of the retrieved key. Additionally, their technique outperforms current frequent pattern mining techniques.

Both unsupervised and supervised key extraction procedures, according to prior debates, have various limitations which prohibit them from getting better outcomes. As a result, as the main feature, this research provides a new unsupervised KCFA technique that will considerably reduce the described defects while also extracting high-quality keyphrases from academic papers.

### 3. Methodology

The proposed KCFA technique has five vital phases (shown in Figure 1): (i) data/datasets Collection, (ii) data pre-processing, (iii) statistical methodologies, (iv) curve plotting analysis, and (v) curve fitting technique. For a better understanding of our proposed technique, we introduce the pseudo code description illustrated in Figure 2. The following subsections provide a more detailed explanation of the proposed system.

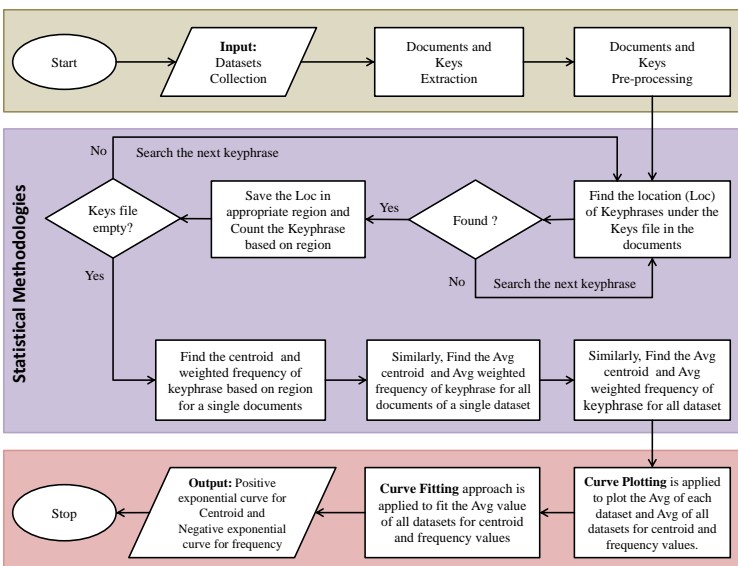

**Figure 1.** The proposed architectural flow diagram for the KCFA technique.

```
1   SET Dataset TO ['citeulike180', 'fao780', 'Krapivin2009',
2       'SemEval2010', 'wiki20', 'PubMed', 'Nguyen2007', '110-PT-
3       BN-KP', 'Theses100', '500N-KPCrowd']
4   SET Markers TO ['*','s','^','o','v','x','<','p','>','d']
5
6   SET X TO [1,2,3,4,5,6,7,8]
7   SET FY, CY TO [], []
8   FOR m IN range(0,len(Dataset),1):
9       SET full_path=[os.path.join(r,file)FOR r,d,f IN
10      os.walk(Datafile[m])FOR file IN f]
11      SET full_path1=[os.path.join(r,file)FOR r,d,f IN
12      os.walk(Keyfile[m])FOR file IN f]
13
14      SET Y2, CY2 TO [], []
15      FOR k IN range(0,len(full_path),1):
16          with open(full_path[k], "r",encoding="utf-8") as f:
17              CALL SET text TO Preprocessing(f.read())
18              SET n TO len(text)
19          with open(full_path1[k], "r",encoding="utf-8") as f:
20              CALL SET key TO Preprocessing(f.read())
21              SET goldkey TO key.split("\n")
22              SET Num_key TO len(goldkey)
23              SET region TO 8
24
25          SET Y1, CY1 TO [], []
26          SET cent TO [[0]*region FOR i IN range(Num_key)]
27          FOR i IN range(Num_key):
28              SET loc TO text.find(goldkey[i])
29              IF (loc >= 0):
30                  FOR j IN range(region):
31                      IF ((loc>=int(j*n/8)) and (loc<int((j+1)*n/8))):
32                          SET cent[i][j] TO (loc+1)
33
34          FOR j IN range(region):
35              SET frq TO 0; Csum1 TO 0
36              FOR i IN range(Num_key): #len(finalkey):
37                  IF cent[i][j]>0:
38                      SET frq TO frq+1
39                      SET Csum1 TO Csum1+cent[i][j]
40              Y1.SAVE(frq*100/Num_key)
41              IF Csum1!=0:
42                  CY1.SAVE(Csum1/frq)
43              ELSE:
44                  CY1.SAVE(Csum1)
45          Y2.SAVE(Y1)
46          CY2.SAVE(CY1)
47
48      SET file_num TO len(full_path)
49      SET Y3, CY3 TO [], []
50      FOR j IN range(region):
51          SET sum2 TO 0; Csum2 TO 0
52          FOR i IN range(file_num): #len(finalkey):
53              SET sum2 TO sum2+Y2[i][j]
54              SET Csum2 TO Csum2+CY2[i][j]
55          Y3.SAVE(round(sum2/file_num,4))
56          CY3.SAVE(round(Csum2/file_num,4))
57      FY.SAVE(Y3)
58      CY.SAVE(CY3
59
60  SET FAvg, CAvg TO [], []
61  SET dataset_num TO len(DatasetName)
62  OUTPUT('dataset_num=',dataset_num)
63

64  FOR j IN range(region):
65      SET sum3 TO 0; Csum3 TO 0
66      FOR i IN range(dataset_num): #len(finalkey):
67          SET sum3 TO sum3+FY[i][j]
68          SET Csum3 TO Csum3+CY[i][j]
69      FAvg.SAVE(round(sum3/dataset_num,4))
70      CAvg.SAVE(round(Csum3/dataset_num,4))
71  OUTPUT("Avg Frequency per Dataset=",FY,'Avg Frequency
72  of all Dataset=',FAvg,)
73  OUTPUT("Avg Centroid per Dataset=",CY,'Avg Centroid of all
74  Dataset=',CAvg,)
75
76  plt.figure()
77  CALL PlottingForCentroid(X,CY,CAvg)
78  CALL PlottingForFrequency(X,FY,FAvg)
79  CALL Curve_Fitting_Centroid(X,CAvg)
80  CALL Curve_Fitting_Frequency(X,FAvg)
81
82  DEFINE FUNCTION Preprocessing(text):
83      SET text TO text.lower()
84      SET text TO re.sub(r'\d+', '', text)
85      SET text TO text.translate(str.maketrans("","",
86      string.punctuation))
87      SET text TO text.strip()
88      RETURN text
89
90  DEFINE FUNCTION PlottingForCentroid(X,CY,CAvg):
91      FOR i IN range(0,dataset_num,1):
92          plt.plot(X,CY[i][0:8], marker=Markers[i],
93      label=DatasetName[i])
94      plt.plot(X,CAvg,'--r', label='Centroid Average')
95      DISPLAY-PLOT()
96
97  DEFINE FUNCTION PlottingForFrequency(X,FY,FAvg):
98      FOR i IN range(0,dataset_num,1):
99          plt.plot(X,FY[i][0:8], marker=Markers[i],
100     label=DatasetName[i])
101     plt.plot(X,FAvg,'--r', label='Frequency Average')
102     DISPLAY-PLOT()
103
104 DEFINE FUNCTION func(x, a, b, c):
105     RETURN a * np.exp(-b * x) + c
106
107 DEFINE FUNCTION Curve_Fitting_Centroid(X,CAvg):
108     SET x TO np.array(X)
109     SET y TO np.array(CAvg)
110     SET yn TO y + 0.2*np.random.normal(size=len(x))
111     CALL SET popt, pcov TO curve_fit(func, x, yn)
112     OUTPUT('Co-efficient of a, b, c=',*popt)
113     plt.plot(x, y, 'ro', label="Average of all datasets")
114     plt.plot(x, func(x, *popt), 'b--', label="Fitted Curve")
115     DISPLAY-PLOT()
116
117 DEFINE FUNCTION Curve_Fitting_Frequency(X,FAvg):
118     SET x TO np.array(X)
119     SET y TO np.array(FAvg)
120     FOR i IN range(len(y)):
121         y[i]=y[i]/100
122     CALL SET popt, pcov TO curve_fit(func, x, y)
123     OUTPUT('Co-efficient of a, b, c=',*popt)
124     plt.plot(x, y, 'ro', label="Average of all datasets")
125     plt.plot(x, func(x, *popt), 'b--', label="Fitted Curve")
126     DISPLAY-PLOT()
```

**Figure 2.** The pseudo code description of the proposed KCFA technique.

### 3.1. Datasets Collection

First, the proposed method collects ten (10) different types of datasets from Github (https://github.com/LIAAD/KeywordExtractor-Datasets (accessed on 20 March 2022)). The datasets are: SemEval2010, citeulike180, fao780, Krapivin2009, Nguyen2007, wiki-20, PubMed, 110-PT-BNKP, theses-100, and 500NKPCrowd, which contain a total of 4948 documents, with two languages (English and Portuguese), four types of documents (including papers, research reports, news, and MSc/PhD theses), and various domains covered (such as computer science, agriculture, and Misce) [43]. Every dataset has two types of files: a keys file (which contains keyphrases) and a documents file, named docsutf8 (which contains the articles or papers). The dataset is explained in detail in Section 4.1.

### 3.2. Data Pre-Processing

When using machine learning techniques, data must be well polished and of high quality in order to yield robust analytical results. To make the original dataset suitable for a high-quality analysis, a number of pre-processing approaches were employed. This phase consists of two processes: documents and keys extraction and documents and keys pre-processing, which are described in the following sub-subsection.

#### 3.2.1. Documents and Keys Extraction

Since every dataset contains two types of files, the proposed approach first extracts both the document files named "docsutf8" (which contain multiple vital documents as text files) and the "keys" files (which contain multiple vital keyphrases as text files). Then, these files are transferred to the documents and keys pre-processing step.

#### 3.2.2. Documents and Keys Pre-Processing

Datasets contain several issues such as punctuation, accent marks and other special characters, numbers, white spaces, abbreviations, uppercase letters, etc. For this reason, they need to be standardized. In our proposed technique, four pre-processing techniques have been utilized, which are: changing the text's case to lowercase; using regular expressions to remove the extraneous numbers; eliminating all punctuation from the text; and removing empty spaces by using the strip() function [44–46]. The split() method is then applied to the key files to calculate the number of keyphrases based on the newline (\n) (also shown in Figure 2). The proposed technique then considers the length of the document is broken into eight (8) regions and employs the first appearance keyphrase to analyze the centroid and frequency of keyphrases.

### 3.3. Statistical Methodologies

This phase consists of two essential processes, such as keyphrase searching, saving, and counting; and keyphrase centroid and weighted-frequency calculation and averaging, described in the following sub-subsections.

#### 3.3.1. Keyphrase Searching, Saving, and Counting

The proposed approach searches for the location (Loc) of the keyphrase under the keys file in the document. If the keyphrase is found in the document, save that location (Loc) of the keyphrase in an appropriate region for further processing and count the keyphrase frequency based on region. Afterward, the proposed technique searches for the next keyphrase in the documents under the keys file if available. If that keyphrase is not found in the documents, the proposed technique is to look for the next keyphrase in the documents. It is important to note that it is saved as a two-dimensional (2D) array, with the document region number in the column and the keyphrase's number in the row. This operation will resume unless keyphrase has finished reading the keys file for one document as well as for a particular dataset. Every dataset will be processed in the same way.

### 3.3.2. Centroid and Weighted-Frequency Calculation and Averaging

The proposed technique gets the keyphrase's locations and the total number of keyphrases present in each region from the earlier processes. First, it computes the centroid value from all the present keyphrase's locations and then calculates the weighted-frequency from the present keyphrase's frequency based on each region of the article. Afterwards, these values need to be saved in two separate 2D arrays, in which the document's number in a dataset is represented as the row and the document's region number is represented as the column [2,4]. The process will then repeat until all of the documents for a given dataset have been completed. Similarly, for each region, calculate the average centroid (AC) value and average weighted frequency (AWF) value of all present keyphrases for all documents in a single dataset, as well as save these values in another two different 2D arrays, in which the dataset's number is indicated as the row and the region number is indicated as the column. The AC and AWF calculation processes will then continue until all datasets have been completed [2]. Finally, the proposed method for every region calculates the AC and AWF values of present keyphrases for all datasets without facing any difficulties. For more details, please visit the Figure 2. The AC and AWF calculation processes of our proposed methodology utilized the following Equations (1) and (2) for each region, respectively. Where $N$, $D$, $T_P$, and $T_K$ denote the number of datasets, documents, total present keyphrase, and total keyphrase. $L$ and $X$ denote the keyphrase location and frequency based on article region.

$$AC = \frac{1}{N}\sum_{i=1}^{N}\left(\frac{1}{D}\sum_{j=1}^{D}\left(\frac{1}{T_P}\sum_{k=1}^{T_P}L_{k,j,i}\right)\right) \tag{1}$$

$$AWF = \frac{1}{N}\sum_{i=1}^{N}\left(\frac{1}{D}\sum_{j=1}^{D}\left(\frac{1}{T_K}\sum_{k=1}^{T_P}X_{k,j,i}\right)\right) \tag{2}$$

### 3.4. Curve Plotting Analysis (CPA)

After the centroid and weighted-frequency calculation and averaging processes, CPA is utilized by the proposed technique. It is a graphical representation technique for all types of values as well as a dataset, and it is pretty beneficial in data statistics and analysis. Curve plotting is utilized to comprehend our proposed KCFA technique based on article region. As a result, the AC value and AWF value of every dataset are plotted separately from the AC value and AWF value of all datasets.

### 3.5. Curve Fitting Technique (CFT)

After the curve plotting analysis process, CFT is used by the proposed technique to analyze nonlinear, linear, and polynomial curves. "This is most probably the method of finding the optimal mathematical equation or curve from a restricted collection of data points" [2]. In our proposed technique, CFT is used to fit the AC value and AWF value for all datasets based upon the article's portion/region as well as to show the average (Avg) centroid and average frequency of keyphrases found in each region. Using our proposed method, CFT produces a positive exponential curve and equation for the keyphrase's centroid and a negative exponential curve and equation for the keyphrase's frequency.

## 4. Experimental Setup

The experimental setting for our proposed approach is fully stated in the following sections. It includes corpus or dataset details, evaluation criteria, and details of implementation.

### 4.1. Corpus Details

The proposed technique uses ten (10) standard datasets to test and evaluate its effectiveness. Another goal was to figure out how the proposed method behaved across a variety of datasets. The ten standard datasets are as follows: citeulike-180, fao-780, Krapivin-2009,

SemEval-2010, wiki-20, PubMed, Nguyen-2007, 110-PT-BN-KP, theses-100, and 500N-KPCrowd [43]. The preceding Section 3.1 has a concise summary of corpus details, and the Table 1 explains a statistical summary of all datasets. This table includes language names, document types, names of domains, number of documents, total keyphrases, total present and absent keyphrases, and the processing time for all datasets. The next paragraph goes through each corpus in great depth.

**Table 1.** Summary of the dataset for analyzing the present-absent keyphrases with processing time.

| Dataset Name | Language | Doc. Types | Domain Name | No. of Docs | No. of Keyphrases | No. of Present Keyphrases | No. of Absent Keyphrase | Processing Time (s) |
|---|---|---|---|---|---|---|---|---|
| citeulike-180 | ENG | Article | Misce. | 183 | 3187 | 2071 | 1116 | 0.531 |
| fao-780 | ENG | Article | Agri. | 779 | 6215 | 3702 | 2513 | 1.860 |
| Krapivin-2009 | ENG | Article | Computer Scien. | 2304 | 12,296 | 9933 | 2363 | 0.984 |
| SemEval-2010 | ENG | Article | Computer Scien. | 243 | 3785 | 3129 | 656 | 1.078 |
| wiki-20 | ENG | Research-Report | Computer Scien. | 20 | 710 | 315 | 395 | 0.016 |
| PubMed | ENG | Article | Life Science | 500 | 7120 | 2513 | 4607 | 0.266 |
| Nguyen-2007 | ENG | Article | Computer Scien. | 209 | 2507 | 2008 | 499 | 0.578 |
| 110-PT-BN-KP | ENG | News | Misce. | 110 | 2688 | 2616 | 72 | 0.047 |
| theses-100 | ENG | MSc/PhD-Thesis | Misce. | 100 | 667 | 302 | 365 | 0.234 |
| 500N-KPCrowd | ENG | News | Misce. | 500 | 24,610 | 22,345 | 2265 | 0.203 |

CiteULike.org is a requirement for the **citeulike-180** corpus, and this collection contains full-text paper documents. Additionally, it is included in the miscellaneous domain, 183 documents, 3187 keyphrases, of which 2071 are available and the remaining are absent, and a processing time of 0.531 s [29].

**fao-780** The collection, which consists of 780 documents, is based on agricultural papers collected from two repositories based on the Food and Agriculture Organization of the United Nations (FAO). It is made up of 780 full-text documents from the FAO collection that were chosen at random. The 6215 keyphrases, of which 3702 are present and 2513 are not, were added by hand by FAO staff using words from the Agrovoc vocabulary [4,29].

The largest collection in terms of the number of documents is **Krapivin-2009**, which contains 2304 complete papers from the Computer Science field that were published in ACM between 2003 and 2005. The papers were acquired from the CiteSeerX Autonomous Digital Library by the authors, who then gave each one a keyword that was later confirmed by the reviewers. It takes 0.984 s to process and has 12296 keyphrases, of which 9933 are present and 2363 are not [29].

The most standardised dataset, **SemEval-2010**, has 244 full scientific papers that were collected from the ACM Library. The articles, which range in length from 6 to 8 pages, cover four different aspects of computer science: distributed artificial intelligence [47], information search and retrieval, social and behavioural sciences, and distributed systems. Each article has a set of keywords chosen by the author and expert editors. It has a processing time of 1.078 s. The present keyphrases of 3129 and 656 are absent [29].

The **wiki-20** collection contains 20 English-language scientific research reports on various topics in computer science. Each report was allocated keywords by 15 teams, each comprised of computer science's two senior students, who used Wikipedia article names as the candidate vocabulary. Each document was to have roughly five keywords assigned to it by the teams. On average, each team submitted 5.7 keywords. It has the present keyphrase of 315 and the absent keyphrase of 395, as well as a processing time of 0.016 s [4].

MIDLINE mentions roughly 0.026 billion ebooks from life science journals in **PubMed** corpuses, which are produced from the PubMed Central full papers. It is a collection of 500 articles culled from the same sources. Medical Subject Headings (MeSH), the constrained keyword dictionary used to expound papers, is the gold keyword in PubMed, with 14.24 keyphrases per document. It has 2513 present and 4607 absent keyphrases, as well as a processing time of 0.266 s [2,4].

**Nguyen-2007**: There are 209 scientific documents in this dataset, together with 2507 keyphrases, of which only 2008 are present. Prior to receiving the keyphrases in person, three papers were given to the student volunteers for review. There are usually 12 keyphrases per article. The processing took 0.578 s [4,29].

"The **110-PT-BN-KP** corpus is a television (TV) broadcast news (BN) corpus that contains 110 transcripts from eight (8) broadcast television news from European Portuguese ALERT BN corpus, including topics such as banking, sports, politics, as well as other topics" [2]. A tagger was used to eliminate all terms that comprised text content summaries, resulting in gold-keys of 24.44 per doc [18]. There are 2616 keyphrases present and 72 keyphrases missing, with a processing time of 0.047 s.

The **Theses-100** corpus contains a hundred entire master's and doctoral theses from New Zealand's University of Waikato. "Computer science, chemistry, economics, philosophy, psychology, and history are just a few of the disciplines represented" [2]. Each document has an average of 6.67 keyphrases. It has 302 available keyphrases and 365 missing keyphrases, with a processing time of 0.234 s [43].

The **500N-KPCrowd** is a dataset of broadcast news transcriptions. This dataset contains 500 English-language broadcast news items, each with 50 documents, from ten distinct groups (crime, art and culture, business, fashion, health, politics, world politics, science, technology, as well as sports). It also contains the present keyphrase of 22,345 and the absent keyphrase of 2265, as well as the processing time of 0.203 s [43,48].

### 4.2. Evaluation Metrics

*Precision, Recall*, and *F1-score* are the three most important and relevant metrics that are used in our proposed technique to compare the performance with other techniques [29,49]. Accuracy by itself is not the optimal measure for imbalanced classes in a dataset since it might be misleading. Once the True Positives and True Negatives are much more significant and the class distribution is similar, accuracy is employed. The proposed technique makes use of the Precession, Recall, and *F1-score* due to the imbalanced classes in our dataset and the importance of the False Negatives and False Positives. The *F1-score* assists in balancing the two measures when a very modest precision or recall will lead to a lower total score. The *F1-score* can help balance the metric between positive and negative samples if we choose the class with fewer samples from the positive class. Here, *Precision* is the ratio of properly predicted values with respect to the total predicted positive values. Another word, it is employed to calculate the positive patterns that are correctly predicted from the total predicted patterns in a positive class. It can be calculated using the following Equation (3):

$$Precision = \frac{Key_{corrected}}{Key_{predicted}} \tag{3}$$

where $Key_{corrected}$ is the total correctly predicted keyphrases that are matched with standard keyphrases and $Key_{predicted}$ is the total predicted keyphrases from a document. On the other hand, *Recall* is the ratio of accurately expected positive values with respect to the actual positive values; and can be calculated using the following Equation (4):

$$Recall = \frac{Key_{corrected}}{Key_{standard}} \tag{4}$$

where $Key_{standard}$ is the total keyphrases in standard keyphrase list for that specific document. Again, *F1-score* is the weighted average of Precision and Recall, which can be

calculated using the following Equation (5). The *F1-score* metric is much more sophisticated than conventional accuracy metric since it takes both false positives and false negatives into consideration.

$$F1 - score = \frac{2 \times Precision \times Recall}{Precision + Recall} \tag{5}$$

### 4.3. Implementation Details

The proposed technique has been developed by utilizing Python 3.6 and the Spyder-IDE. This is an object-oriented, high-level programming language that is also very simple-to-learn and use. It offers a flexible, user-friendly data format that is also supported by a wide range of libraries. It is open-source and free, increases productivity, and is interpretative and dynamically typed. It is employed in a variety of domains, including big data, machine learning, and cloud computing. The laptop is then equipped with just an Intel Core-i7 processor, a 256 GB of SSD-drive, 12 GB of RAM, and Windows-10 OS [2,29].

## 5. Results and Discussion

This section carefully examines the experiment's findings. The proposed method splits the document length into eight (8) regions to analyze the keyphrase centroid and frequency (KCF). When a document's number of regions is increased by more than eight, the first region has a lower keyphrase frequency value as well as a lower centroid value than the eight-region. Likewise, if the region numbers are fewer than eight regions, the first area or region has a higher keyphrase frequency value as well as a higher centroid value than the eight regions. Since the proposed method aims to show the centroid and frequency of keyphrases from articles based on each region, the model considers the article length in eight regions instead of expanding or reducing the area/region. The two most important parts of this section are *analyzing the results* and *comparing the proposed method* to other methods. These two parts are explained in the next subsection.

### 5.1. Results Analysis

The performance of the proposed technique is assessed by utilizing the following three forms of analysis: first datasets/corpus analysis, second plotting analysis, and finally curve fitting analysis.

#### 5.1.1. Dataset Analysis

The proposed method is tested using ten (10) standard datasets (see in detail in Section 4.1). The proposed technique then counts the number of documents (doc), the number of keyphrases, the number of present keyphrases, and the number of absent keyphrases as well as calculates the processing time (sec) per dataset shown in Table 1. The average number of present and absent keyphrases in each doc is analyzed for every dataset and is demonstrated in Figure 3. Similarly, the average present keyphrases as well as absent keyphrases per document (%) for each dataset are illustrated in Figure 4. The proposed technique's results demonstrate that, on average, 68.11% of keyphrases are present per document across all datasets, while 31.89% of keyphrases are either missing or absent.

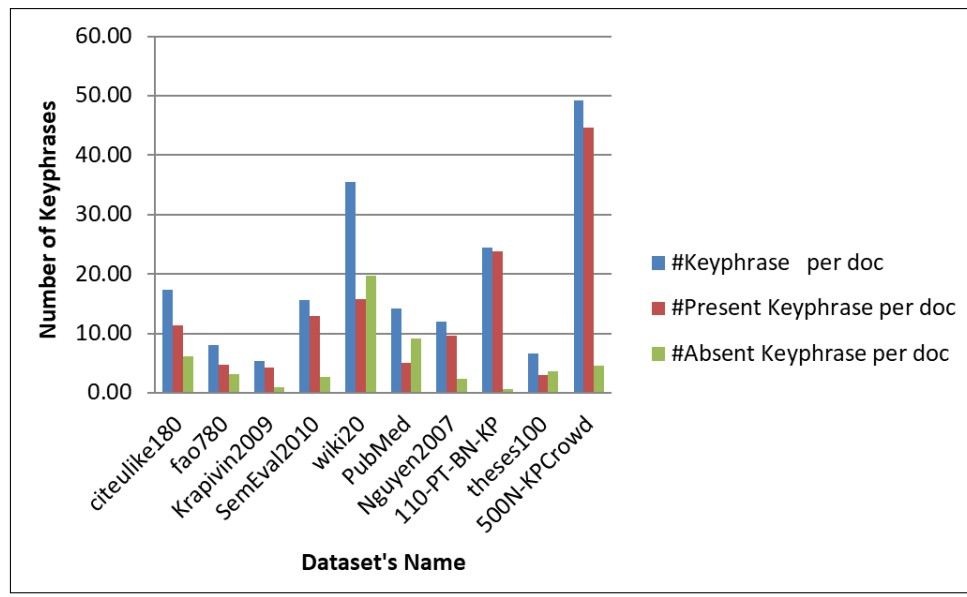

**Figure 3.** Analyze the present and absent keyphrases per doc for every dataset.

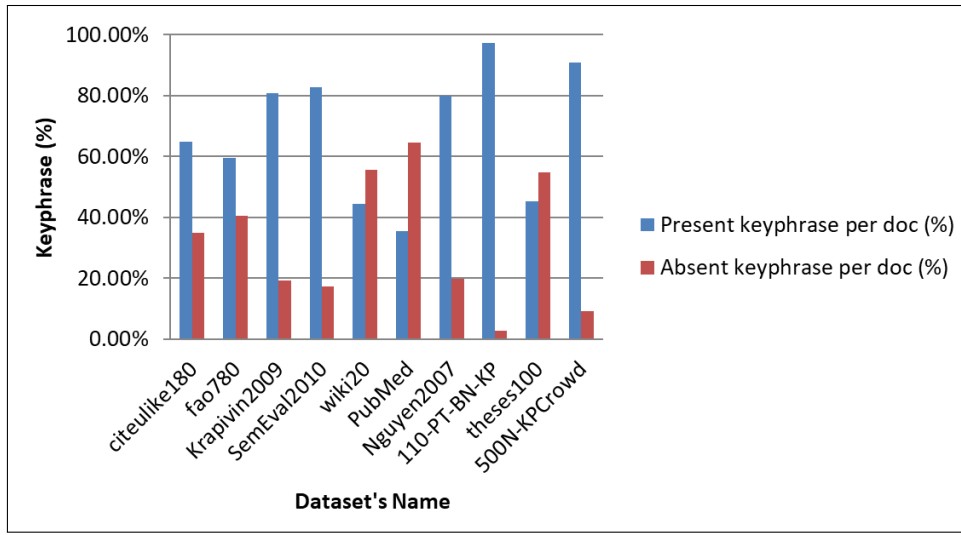

**Figure 4.** Analyze the Avg percentage (%) of present and absent keyphrases per doc for all datasets.

5.1.2. Plotting Analysis

Since the proposed technique finds an Avg of 68.11% of keyphrases are present per doc for all datasets, all outcomes are focused on 68.11% of keyphrases. First, the proposed KCFA technique plots the Avg values of each dataset together, and then the Avg values of all datasets based on each article region. For the centroid analysis of keyphrases in per region, the proposed technique plots the 1st five dataset's Avg centroid (AC) value and then plots the 2nd five dataset's AC value separately for clear visualization as shown in Figures 5 and 6.

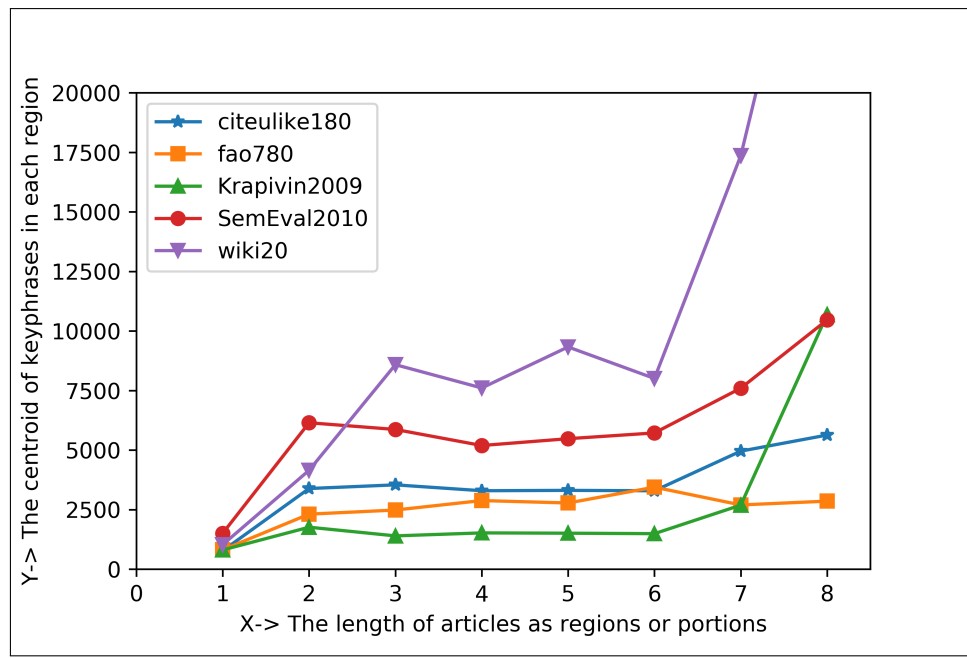

**Figure 5.** The analysis of keyphrase centroid using first occurrence keyphrase based on eight regions for 1st five datasets.

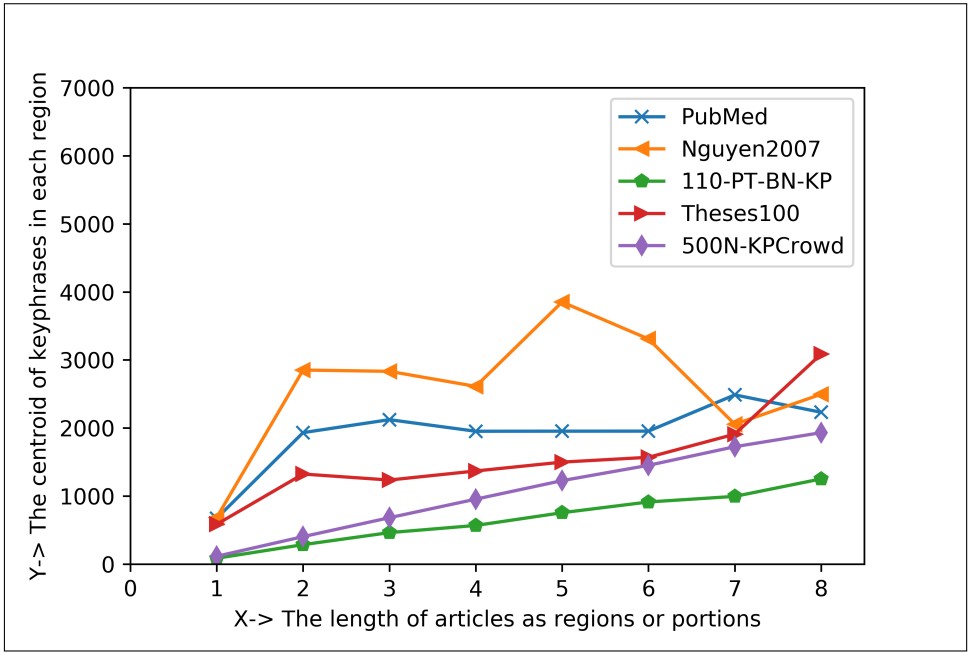

**Figure 6.** The analysis of keyphrase centroid using first occurrence keyphrase based on eight regions for 2nd five datasets.

Similarly, for the frequency analysis of keyphrases in percentage per region, the proposed method plots the 1st five dataset's Avg weighted frequency (AWF) value and then plots the 2nd five dataset's AWF value separately for the same consideration exhibited in Figures 7 and 8.

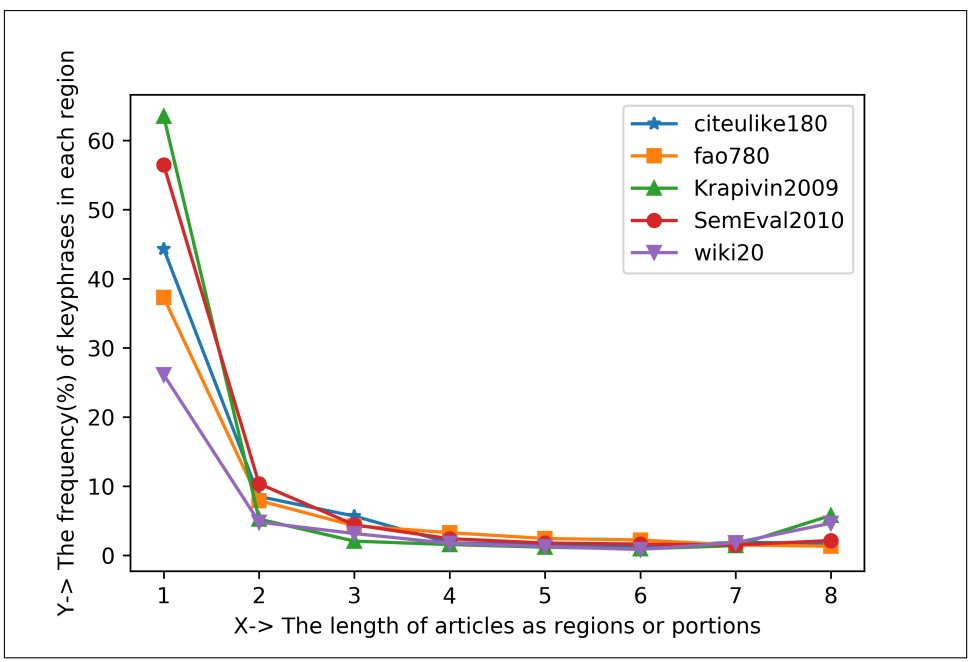

**Figure 7.** The analysis of keyphrase frequency in percent (%) using first occurrence keyphrase based on eight regions for 1st five datasets.

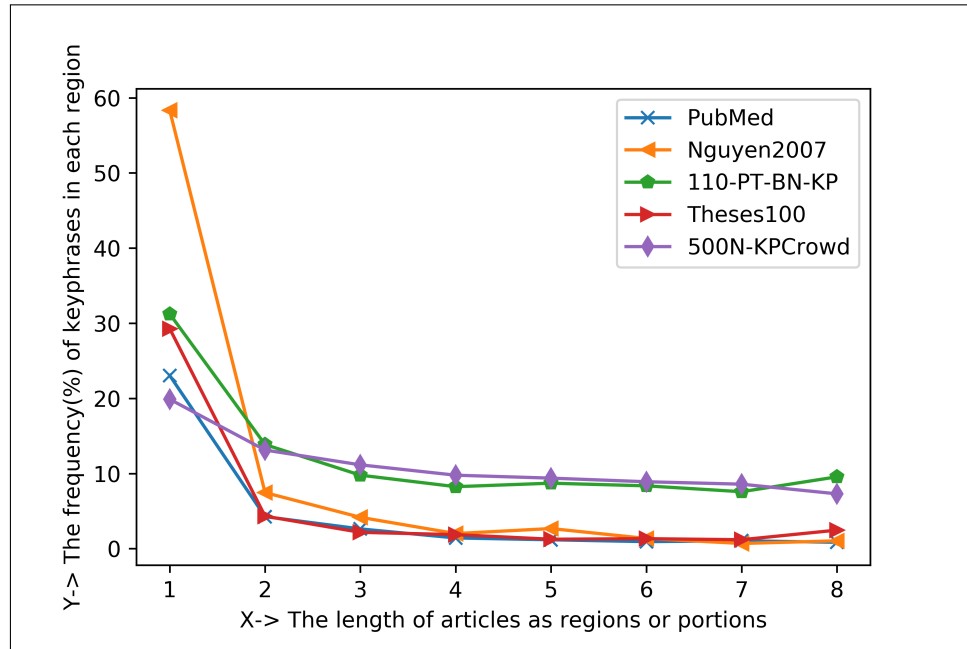

**Figure 8.** The analysis of keyphrase frequency in percent (%) using first occurrence keyphrase based on eight regions for 2nd five datasets.

Once again, the proposed KCFA technique plots the AC value of all datasets collectively to illustrate the analysis of the Avg keyphrase centroid based on each region, as shown in Figure 9. Similarly, Figure 10 shows the analysis of the Avg keyphrase weighted frequency in percent (%) based on each region under the same consideration as before. Because all of the dataset curves for keyphrase centroid analysis are positively exponential, it is evident that the lowest keyphrase centroid value is found in the first region of an article, followed by greater values in the second region, and so on, as shown in Figures 5, 6 and 9. Similarly, since the proposed technique found negative exponential curves for the frequency analysis of keyphrases, it is demonstrated that the highest keyphrase frequencies are found

in the first region, followed by lower keyphrase frequencies in the second region, and so on, as shown in Figures 7, 8 and 10.

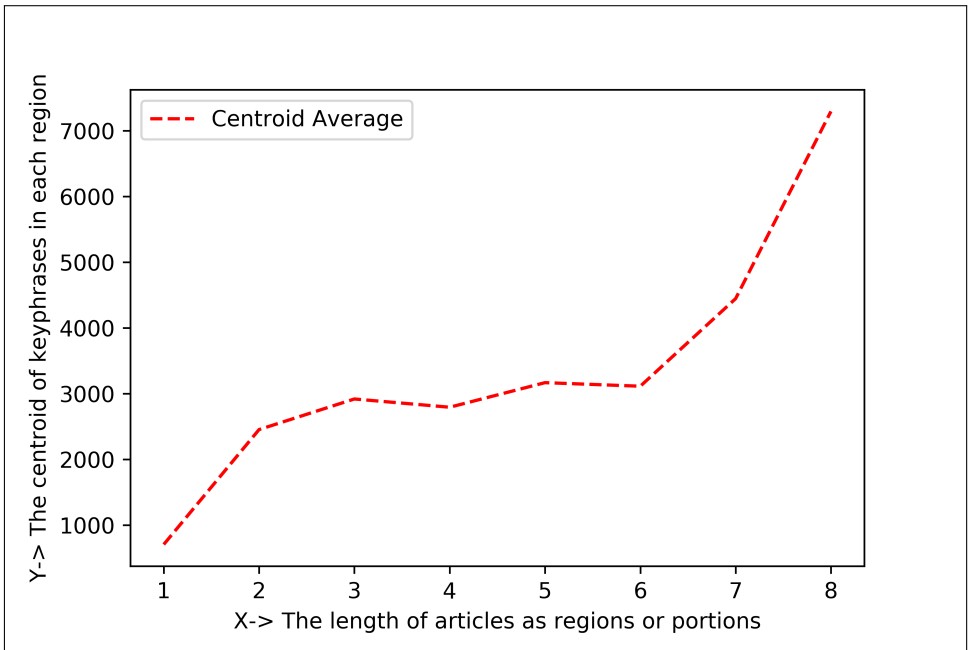

**Figure 9.** The analysis of Avg centroid value of all datasets using the same consideration for KCFA technique.

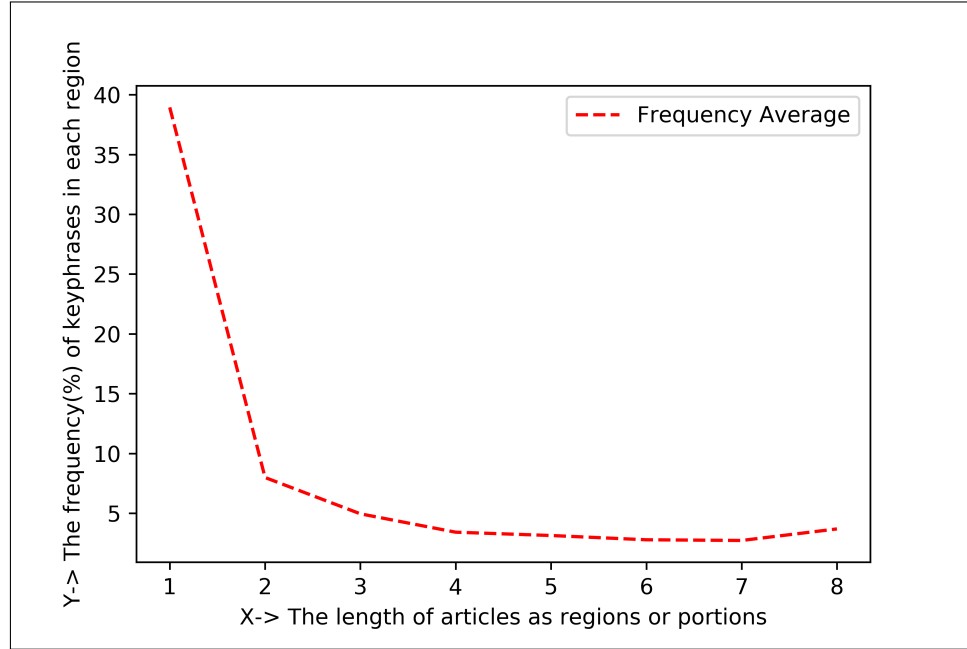

**Figure 10.** The analysis of Avg weighted frequency of all datasets in percent (%) using same consideration for KCFA technique.

### 5.1.3. Curve Fitting Analysis

After the plotting analysis, the proposed KCFA method uses curve fitting to fit the AC and AWF values of all datasets and measure how well the method works. Afterwards, the proposed technique discovered the positive exponential curve and equation for the AC value for every region. The analysis of the curve fitting approach for the AC value of all datasets

based on 8 regions for the proposed KCFA technique is exhibited in Figure 11, and also provides a positive exponential Equation (6), where $p = 47.16$, $q = 0.59$, and $r = 1916.63$.

$$y = p * e^{+qx} + r \tag{6}$$

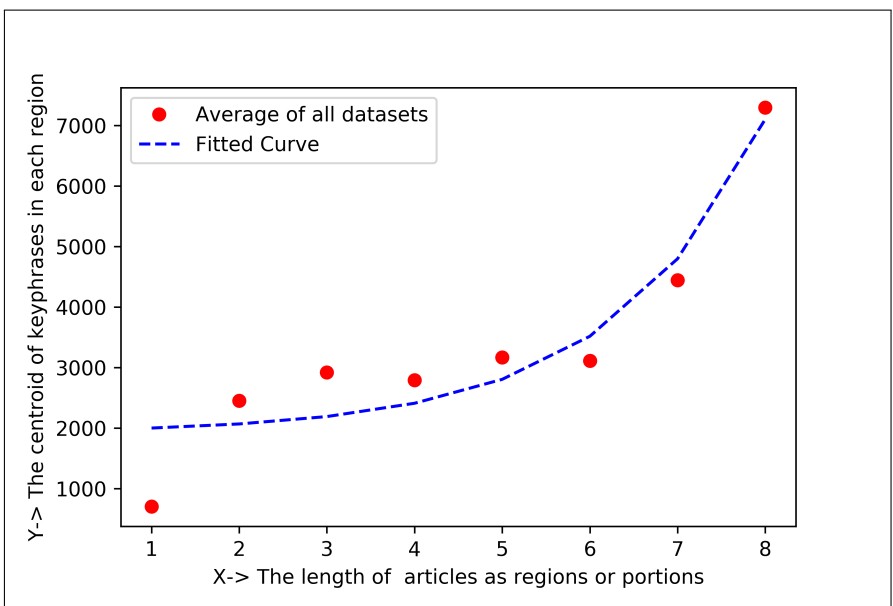

**Figure 11.** The analysis of curve fitting technique for AC value of all datasets based on 8-regions.

Similarly, the proposed technique then discovered the negative exponential curve/equation for the AWF value by utilizing this technique. The analysis of the curve fitting approach for the AWF value of all datasets based on each region of the proposed KCFA technique is shown in Figure 12, with the same consideration. It also offers a negative exponential Equation (7), where $m = 2.55$, $n = 1.97$, and $l = 0.03$.

$$y = m * e^{-nx} + l \tag{7}$$

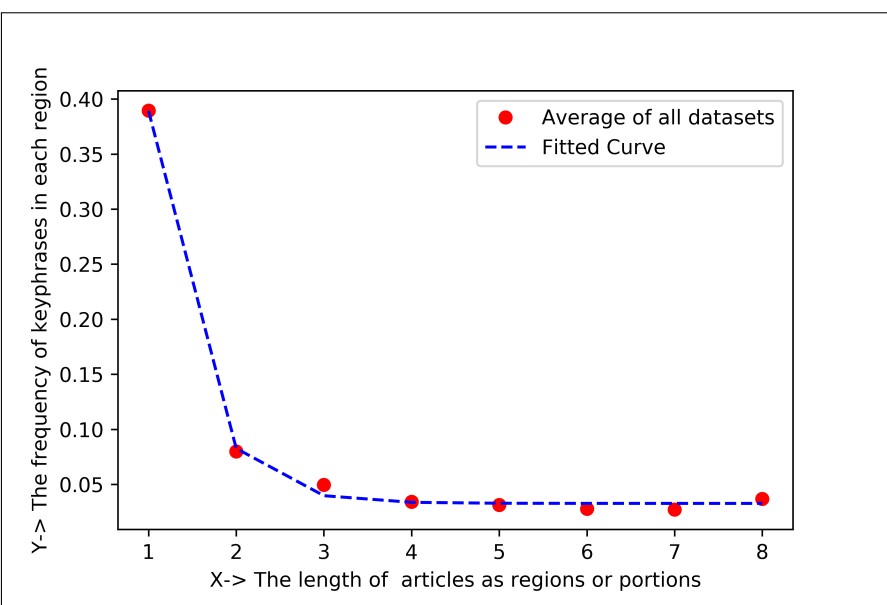

**Figure 12.** The analysis of curve fitting approach for the AWF value of all datasets based based on 8-regions.

From the curve fitting analysis, it is proven that the maximum keyphrase frequency value is found with a lower centroid value in the first region of articles, then in the second region, and so forth, as illustrated in Figures 11 and 12, respectively. This is because the proposed approach receives the negative exponential curve/equation for the keyphrase frequency as well as the positive exponential curve/equation for the keyphrase centroid. Lastly, the proposed technique can also be summarized from Figures 9–12, where the keyphrase frequency is found to be 38.95% with a centroid value of 706.66 in the first region, 7.98% with a centroid value of 2454.21 in the second region, and so forth, where the present keyphrase frequency is 68.11% and the absent keyphrases are 31.89%.

*5.2. Comparison of Proposed Systems*

The proposed method has been used two types of comparisons. First, it compares the performance of all the datasets to choose a better dataset, and second, it compares with the existing technique to choose a better model or technique; these are detailed in the next subsection.

### 5.2.1. Comparison for Choosing a Better Dataset

The proposed technique uses the evaluation metrics (such as precision, recall, and f1-score) to measure the performance of each dataset and find a better one. Since the proposed KCFA technique is a keyphrase centroid and frequency analysis technique and not a keyphrase extraction technique, if it has no Actual Negative value, this indicates that if the key is present in the documents, the proposed technique can easily find it. Similarly, if the keyphrase is absent from the documents, the technique cannot find that keyphrase. For this reason, the *Precision* value in our proposed approach is always 100%, and also because $Key_{corrected}$ and $Key_{predicted}$ are always the same. The performance comparison of all datasets to find a better dataset is shown in Table 2. In this table, we can find that the "110-PT-BN-KP" dataset provides the 1st highest *F1-score* of 98.64%, the "500N-KPCrowd" dataset provides the 2nd highest *F1-score* of 95.18%, and the "SemEval2010" dataset provides the 3rd highest *F1-score* of 90.51%. Finally, it is demonstrated that the "110-PT-BN-KP" dataset is better than other datasets in our proposed approach.

**Table 2.** Performance comparison of all datasets for finding a better one.

| Dataset's Name | Performance Measurements | | |
| | Precision | Recall | F1-Score |
|---|---|---|---|
| citeulike180 | 100.00% | 64.98% | 78.78% |
| fao780 | 100.00% | 59.57% | 74.66% |
| Krapivin2009 | 100.00% | 80.78% | 89.37% |
| **SemEval2010** | **100.00%** | **82.67%** | **90.51%** |
| wiki20 | 100.00% | 44.37% | 61.46% |
| PubMed | 100.00% | 35.29% | 52.17% |
| Nguyen2007 | 100.00% | 80.10% | 88.95% |
| **110-PT-BN-KP** | **100.00%** | **97.32%** | **98.64%** |
| theses100 | 100.00% | 45.28% | 62.33% |
| **500N-KPCrowd** | **100.00%** | **90.80%** | **95.18%** |

### 5.2.2. Comparison for Choosing a Better Technique

Since the proposed KCFA technique is a new technique and there is no existing technique, it cannot be compared to other techniques. For this reason, it compares with our two suggested approaches, such as eight (8) regions and sixteen (16) regions, as indicated in Table 3. For both approaches, the proposed technique uses the same ten (10) standard datasets for comparison purposes to show which one is better. According to Table 3, the proposed technique can show that the average keyphrase frequency value as well as its centroid value in each region of the eight-region approach is higher than the sixteen-region approach. These two approaches also prove that the proposed KCFA technique for analyzing the center and frequency of keyphrases in the articles is correct.

**Table 3.** Comparison of our proposed two approaches for KCFA technique.

| Articles Regions | Keyphrase Centroid and Frequency (%) in 1st Region | Keyphrase Centroid and Frequency (%) in 2nd Region | Co-Efficient for Positive Exponential ($p*e^{+qx}+r$) | Co-Efficient for Negative Exponential ($m*e^{-nx}+l$) |
|---|---|---|---|---|
| Eight Regions | 706.66 and 38.95% | 2454.21 and 7.98% | $p = 47.16, q = 0.59,$ $r = 1916.63$ | $m = 2.55, n = 1.97,$ $l = 0.033$ |
| Sixteen Regions | 386.69 and 31.89% | 1223.87 and 7.05% | $p = 3.97, q = 0.43,$ $r = 1378.87$ | $m = 1.50, n = 1.62,$ $l = 0.019$ |

## 6. Conclusions

This paper introduces KCFA, a novel unsupervised region-based approach for analyzing keyphrases' centroid and frequency from articles. It is also independent of domain as well as language, requires little statistical understanding, but does not rely on training data. The proposed approach begins with data acquisition and then pre-processing, then moves on to statistical methodologies, curve plotting analyses, and lastly, the curve fitting procedure. Afterwards, the proposed technique was tested and validated on ten (10) standard datasets to measure the performance and produce a positive exponential curve and equation for the centroid value and a negative exponential curve and equation for frequency, indicating that most of the keyphrase frequencies with their centroid value are located in the first region of articles, then the second region, and so forth. Finally, it is noted that the proposed KCFA technique effectively analyzes the keyphrase centroid and frequency from the articles based on region and delivers 706.66 and 38.95% of the keyphrase centroid and keyphrase frequency, respectively, in the 1st region, 2454.21 and 7.98% in the 2nd region, and so on, where a total keyphrase frequency of 68.11% is present and 31.89% is absent for all datasets. The proposed approach also finds the best dataset named "110-PT-BN-KP" with the highest accuracy of 97.32% and the highest *F1-score* of 98.64% as well as it will improve the effectiveness of present keyphrase extraction methods significantly. The demerits of the proposed technique are that it takes more processing time for long-length articles and it only measures the keyphrase centroid and frequency of any article. In the future, we hope to design a robust keyphrase extraction algorithm, making use of the statistical data gathered during this study. We are also working on a solution for the problem of missing keyphrases, which happens when several manually assigned keyphrases are not found in the text.

**Author Contributions:** Conceptualization, M.B.A.M. and S.A.; methodology, M.B.A.M. and S.A.; software, M.B.A.M.; validation, M.M.R. and S.A.; formal analysis, M.B.A.M.; investigation, S.A.; resources, A.S.M.S.H. and I.-H.R.; data curation, M.B.A.M.; writing—original draft preparation, M.B.A.M.; writing—review and editing, M.M.R. and S.A.; visualization, M.B.A.M.; supervision, S.A.; project administration, S.A.; funding acquisition, A.S.M.S.H. and I.-H.R. All authors have read and agreed to the published version of the manuscript.

**Funding:** This research was funded by the Universiti Malaysia Pahang (UMP) through the FLAGSHIP Research Scheme under Grants (RDU192210 and RDU192212) and The APC was fully funded by the National Research Foundation of Korea (NRF) grant by the Korean Government through the Ministry of Science and ICT (MSIT) under Grant 2021R1A2C2014333.

**Institutional Review Board Statement:** Not applicable.

**Informed Consent Statement:** Not applicable.

**Data Availability Statement:** The dataset as well as code of this study are available on request from the 1st author or The dataset presented in this study are available at KeywordExtractor-Datasets repository.

**Acknowledgments:** The authors gratefully acknowledge the Universiti Malaysia Pahang (UMP) for providing laboratory space and financing, as well as the National Research Foundation of Korea (NRF).

**Conflicts of Interest:** The authors declare no conflict of interest.

**Abbreviations**

The following abbreviations are used in this manuscript:

| | |
|---|---|
| KCFA | Keyphrases Centroid and Frequency Analysis |
| SML | Supervised Machine Learning |
| NLP | Natural Language Processing |
| KCF | Keyphrase Centroid and Frequency |
| CPA | Curve Plotting Analysis |
| CFT | Cure Fitting Technique |
| TeKET | Tree-based Keyphrase Extraction Technique |
| YAKE | Yet Another Keyword Extractor |
| TF-IDF | Term Frequency–Inverse Document Frequency |
| NN | Neural Networks |
| SVM | Support Vector Machines |
| DT | Decision Trees |
| KEA | Keyphrase Extraction Algorithm |
| GenEx | Genitor Extractor |
| POS | Part-of-speech |
| DPM | Document Phrase Maximality |
| CeKE | Citation-enhanced Keyphrase Extraction |
| AWF | Average Weighted Frequency |
| AC | average centroid |
| Loc | Location |
| Avg | Average |
| GRISP | General Research Insight in Scientific and technical Publications |
| HAL | Hyper Article en Ligne |
| GROBID | GeneRation Of BIbilographic Data |
| TEI | Text Encoding Initiative |

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
