# Peer review of "A New Unsupervised Technique to Analyze the Centroid and Frequency of Keyphrases from Academic Articles"

_electronics, doi:10.3390/electronics11172773_

Round 1

Reviewer 1 Report

However, there are still several limitations regarding the paper quality.

1. Paper motivation can be further improved by adding an intuitive example.

2. Why do you choose Precision, Recall and F1-score as the evaluation metrics in the experiments? More discusssions are necessary.

3. In the experiment evaluation section, the authors introduce several compared approaches. The authors should explain why they select these approahches for comparisons.

4. Paper formulation needs to be refined by introducing pseudo code description about the introduced proposal.

5. Natural language processing technique studied in this paper has been investigated for long decades and there are already a number related literatures focusing on the same topic. Therefore, I suggest the authors to introduce the following related papers in revision: Pretrained Models and Evaluation Data for the Khmer Language; From Symbols to Embeddings: A Tale of Two Representations in Computational Social Science; Text-Based Price Recommendation System for Online Rental Houses; CDCAT: A Multi-Language Cross-Document Entity and Event Coreference Annotation Tool.

Reviewer 2 Report

This manuscript proposes a novel unsupervised region-based key-phrase centroid and frequency analysis (KCFA) technique to analyze the centroid and frequency of key-phrase, for the key-phrase extraction technique as a feature. Extensive empirical studies on the ten best accessible benchmark datasets in comparison show the effectiveness of the proposed method. However, this paper is not well written, and its technical details are hard to follow, the following issues need to be further revised and improved:

1. Abbreviations should be explained when they are used first in the text, such as “KCFA” in Abstract, “HUMB” in line 114, page 3, and “GRISP, HAL, and GROBID/TEI” in line 117, page 3.

2. This paper lacks comparative experiments to show the effectiveness of the experiment. The models of recent years should be introduced as the benchmark in the experiment, and the references of these comparative models should be cited.

3. It would be better to discuss the limitations of the proposed method in the conclusion.

4. Presentation issue. Authors should use more academic language to illustrate this paper to further improve its readability. For instance, “in this phase” in line 199, page 5, “Sometimes” in line 204, page 5, and “After that” in line 209, page 6. There are too many redundant statements in the text, “this phase is very important compared to the other phases…” in line 231, page 6, “CPA is a very important step for our proposed technique.” in line 251, page 7.

5. There are some grammatical errors in this manuscript, for instance, --“In our recommended technique utilizes the ten…” in line 274, page 7, --"while Table 1 contains…” in line 277, page 7. Improper use of singular and plural forms in the manuscript, for instance, “row” in line 237, and “process” in line 246.

Please the authors have an overall check to eliminate all similar mistakes in the whole manuscript.

6. What the “Newline (backslash) method” is? Please provide references.

7. The layout of Table I needs to be improved. Fig. 1 and Fig. 2 are more suitable for using histograms with digital labels.

8. Literature review. Many cited papers are outdated, such as [5,7,10,12,13,15-17,22,25,27,31,33,36,41-44]. To introduce the latest studies to the readers, please cite more recent related studies. Such as the following related studies.

[1].    Liu, F., Zhang, G., & Lu, J. (2020). Multi-source heterogeneous unsupervised domain adaptation via fuzzy-relation neural networks. IEEE transactions on fuzzy systems, 1. doi: 10.1109/TFUZZ.2020.3018191

[2].    Zhong, L., Fang, Z., Liu, F., Yuan, B., Zhang, G.,... Lu, J. (2021). Bridging the Theoretical Bound and Deep Algorithms for Open Set Domain Adaptation. IEEE transaction on neural networks and learning systems, PP, 1-15. doi: 10.1109/TNNLS.2021.3119965

[3].    Wu, X., Zheng, W., Xia, X., & Lo, D. (2021). Data Quality Matters: A Case Study on Data Label Correctness for Security Bug Report Prediction. IEEE transactions on software engineering, 1. doi: 10.1109/TSE.2021.3063727

[4].    Zheng, W., Xun, Y., Wu, X., Deng, Z., Chen, X.,... Sui, Y. (2021). A Comparative Study of Class Rebalancing Methods for Security Bug Report Classification. IEEE transactions on reliability, 70(4), 1-13. doi: 10.1109/TR.2021.3118026

[5].    Zhang, M., Chen, Y., & Susilo, W. (2020). PPO-CPQ: A Privacy-Preserving Optimization of Clinical Pathway Query for E-Healthcare Systems. IEEE internet of things journal, 7(10), 10660-10672. doi: 10.1109/JIOT.2020.3007518

[6].    Zheng, W., Zhou, Y., Liu, S., Tian, J., Yang, B.,... Yin, L. (2022). A Deep Fusion Matching Network Semantic Reasoning Model. Applied Sciences, 12(7), 3416. doi: 10.3390/app12073416

[7].    Di Wu, Yi He, Xin Luo, and MengChu Zhou, A Latent Factor Analysis-based Approach to Online Sparse Streaming Feature Selection, IEEE Transactions on Systems Man and Cybernetics-Systems, 2021, DOI: 10.1109/TSMC.2021.3096065

[8].    Zheng, W., & Yin, L. (2022). Characterization inference based on joint-optimization of multi-layer semantics and deep fusion matching network. PeerJ Computer Science. doi: 10.7717/peerj-cs.908

[9].    Zheng, W., Tian, X., Yang, B., Liu, S., Ding, Y., Tian, J.,... Yin, L. (2022). A Few Shot Classification Methods Based on Multiscale Relational Networks. Applied Sciences, 12(8). doi: 10.3390/app12084059

Reviewer 3 Report

1-   Remove  KCFA  from  Tittle,  authors should put full name 

2- The abstract is not clear, why we need this study?  authors need  to rewrite the abstract  and  give clear  centroid. 

3-  The  formwork is  not clear, the proposed architectural flow diagram for the KCFA technique.,  authors  should  shows  the flowchart of algorithm 

how   you use the centroid of clustering approaches for developing your study.

4- This is very basic article,   and this  article is not appropriate to electronic journal, authors should  find  education journal  for submitting   this article 

Round 2

Reviewer 2 Report

All my concerns have been addressed. This paper could be accepted for publication.

Reviewer 3 Report

Authors  have been addressed all comments